# The Multifaceted Relationship between the COVID-19 Pandemic and the Food System

**DOI:** 10.3390/foods11182816

**Published:** 2022-09-13

**Authors:** Antonello Paparella, Chiara Purgatorio, Clemencia Chaves-López, Chiara Rossi, Annalisa Serio

**Affiliations:** Faculty of Bioscience and Technology for Food, Agriculture and Environment, University of Teramo, 64100 Teramo, Italy

**Keywords:** SARS-CoV-2, COVID-19, food transmission, food safety, persistence, food system

## Abstract

The SARS-CoV-2 pandemic is being questioned for its possible food transmission, due to several reports of the virus on food, outbreaks developed in food companies, as well as its origins linked to the wet market of Wuhan, China. The purpose of this review is to analyze the scientific evidence gathered so far on the relationship between food and the pandemic, considering all aspects of the food system that can be involved. The collected data indicate that there is no evidence that foods represent a risk for the transmission of SARS-CoV-2. In fact, even if the virus can persist on food surfaces, there are currently no proven cases of infection from food. Moreover, the pandemic showed to have deeply influenced the eating habits of consumers and their purchasing methods, but also to have enhanced food waste and poverty. Another important finding is the role of meat processing plants as suitable environments for the onset of outbreaks. Lessons learned from the pandemic include the correct management of spaces, food hygiene education for both food workers and common people, the enhancement of alternative commercial channels, the reorganization of food activities, in particular wet markets, and intensive farming, following correct hygiene practices. All these outcomes lead to another crucial lesson, which is the importance of the resilience of the food system. These lessons should be assimilated to deal with the present pandemic and possible future emergencies. Future research directions include further investigation of the factors linked to the food system that can favor the emergence of viruses, and of innovative technologies that can reduce viral transmission.

## 1. Introduction

COVID-19 is a respiratory disease caused by a new type of coronavirus, the SARS-CoV-2 (Severe Acute Respiratory Syndrome Coronavirus-2). Coronaviridae is a large family of enveloped, positive-sense single-stranded RNA viruses that includes many strains [1]. However, the provisionally called 2019 novel coronavirus (2019-nCoV), now known as SARS-CoV-2, was first reported to infect humans in Wuhan, Hubei Province, China, in December 2019 [2,3]. The World Health Organization (WHO) declared the outbreak a Public Health Emergency of International Concern on 30 January 2020, and a pandemic on 11 March 2020 [4,5]. Although the exact origin of the virus is still unknown, the most likely natural hosts of SARS-CoV-2 are bats, because of the high homology in their nucleotide sequence (96%) at the whole genome level [6,7]. However, it is believed that the virus jumped from species to species, passing to other domestic or wild animals, and finally to humans [8], with pangolin as a probable intermediate host [9].

The periodic appearance of coronaviruses in humans is very likely, especially in situations where people and animals are in close contact [10]. In fact, time after time, the invasion of the animal habitat by humans determined the appearance of zoonoses [11]. In this context, an important role might have been played by the food system. In fact, many factors, such as direct or indirect contact with animals, especially wild species, or the consumption of their meat, particularly inadequately cooked, could have determined the spillover that leads to the emergence of a new zoonotic disease [12]. For SARS-CoV-2, the first evidence is most likely to be traced back to the Huanan seafood wholesale market (Wuhan, China), a wet market selling both food and live wild animals [13,14,15]. However, there are no reports of which live animals were sold in the months preceding the pandemic, making it difficult to understand which species were involved in the spillover [15]. Despite this, several pieces of evidence support the fact that the Huanan market was the epicenter of the pandemic. For example, environmental analyses within the market showed that SARS-CoV-2 positive samples were associated with areas where live animal sales were practiced, just before the start of the pandemic [15]. Another key consideration is that among the early COVID-19 patients, those who did not work or had not been in contact with market workers or had not visited the market, lived closer to the market than those who worked there, thus demonstrating exposure to the virus not due to their work, but to the proximity of their residence to the market [15].

Due to the origin and implications of SARS-CoV-2 in the food system, the role of food in the transmission of the virus has been hypothesized. Concerns are mainly related to the possible contamination of food products, especially raw food, from handling by operators who contracted the virus, or directly from intermediate domestic host animals. Food could represent a vehicle of SARS-CoV-2 during the various stages of processing, from production to sale, because the virus is able to survive on organic tissues for a long time. Furthermore, SARS-CoV-2 can persist for prolonged periods even on inorganic surfaces, such as those of food packaging or work surfaces of food companies. Even these fomites, contaminated by infected workers or by the food itself, are thought to play a role in transmitting the virus. To reinforce this hypothesis, there are several outbreaks reported in food companies. To date, as evidenced by the assessments of multiple national and international bodies operating in the field of food safety, and namely WHO, EFSA (European Food Safety Authority), FDA (Food and Drug Administration), and CDC (Centers for Disease Control and Prevention), there are no known cases of SARS-CoV-2 transmission through foods, and the risk is remote [16,17,18,19,20]. Despite this, the belief that food can transmit COVID-19 is still present, and the role of food in the transmission of the virus cannot be categorically excluded.

Nevertheless, the lack of evidence on a link between SARS-CoV-2 and food transmission is consistent with the experience of previous coronavirus infections. In fact, the virus needs an animal or human host to grow, and cannot grow in food, as for bacteria. For this reason, a food worker is much more likely to be infected by person-to-person transmission than through contaminated food. As a matter of fact, COVID-19 is generally considered a problem that concerns the health sector rather than food contamination [21,22]. However, the possibility that it can survive on food, especially raw food, and packaging is real, and therefore it is deemed necessary to perform good hygiene practices to avoid any kind of risk, albeit remote, of infection through food [23,24]. In this respect, WHO advised cooking food properly, avoiding cross-contamination between cooked and raw food, washing hands often and carefully, covering mouth when coughing or sneezing, avoiding preparing food for other people in the presence of symptoms related to COVID-19, sanitizing the environments in which foods are prepared [8]. In general, adequate cooking of food is very important, because raw or undercooked meat, as well as other fresh foods, could also carry many other pathogenic microorganisms [25]. In particular, wild-hunted game meat could be the vector of other unknown microorganisms that can determine new zoonoses [26]. Since the investigation of the potential relationships between SARS-CoV-2 and food is an interdisciplinary topic, this review aims to comprehensively describe the multifaceted relationship between the COVID-19 pandemic and the food system.

## 2. Materials and Methods

We conducted the research following specific steps. First, we identified the key points to be addressed: the persistence and detection of SARS-CoV-2 on foods, packaging, and work surfaces, the impact of the pandemic on the food system (home environment, commercial activities, food access, food loss, and food waste), the role of food manufacturing in the onset of specific outbreaks, guidelines, and recommendations for the food sector, lessons learned, and research needs for the future. Afterward, we consulted different search engines (Google Scholar, PubMed) and different portals of national and international organizations (WHO, EFSA, CDC, FDA), up to 24 August 2022, in which we entered keywords (“SARS-CoV-2”, “COVID-19”, “virus”, “coronavirus”, “pandemic”, “food”, “food system”, “persistence”, “detection”, “food packaging”, “food surface”, “work surface”, “impact”, “home environment”, “commercial activities”, “food access”, “food loss”, “food waste”, “food companies”, “outbreak”, “slaughterhouse”, “guidelines”, “recommendations”, “resilience”, etc., plus a combination of several words). Then, we analyzed and performed a skimming of the current literature regarding each of the above points. We reported the most significant evidence for the topics, discussing and comparing each aspect. Finally, the conclusions report the major findings and potential outcomes of the work. We found a relevant study using a hierarchical approach based on title, abstract, and then the entire manuscript. We consulted articles, reviews, official guidelines, and scientific opinions, and we also used references from the extracted works. When multiple articles for a single aspect were present, we used the latest and more complete and updated publications.

## 3. Persistence, Detection, and Disinfection Methods of SARS-CoV-2 in the Food System

### 3.1. Persistence on Food

In theory, the possibility of food transmission of SARS-CoV-2 derives from the ability of the virus to survive on different types of surfaces. Indeed, it is shown that the stability of this virus on surfaces is in general lower than that of many other pathogens, such as various non-enveloped viruses or bacterial spores [27]. The factors that influence the survival of the virus on surfaces are temperature, humidity, daylight, surface conditions, and material, but also the variant and the viral load.

SARS-CoV-2, like other human coronaviruses, survives more on wet surfaces than on dry surfaces [28]. In dry conditions, the virus can survive within a period from a few hours to a maximum of two days [29]. Sunlight and high temperatures can favor the inactivation of the virus. In fact, as already known for other viruses, SARS-CoV-2 can remain more infectious at low temperatures [30,31], also on food surfaces such as meat or fish tissues. A factor that can contribute to the persistence of the virus on meat is that meat from many animals, such as beef, pork, poultry, and wild animals, is rich in heparin sulfate, which is needed by SARS-CoV-2 to adhere to the host epithelium [32]. Fisher et al., (2020) studied the persistence of SARS-CoV-2 on frozen meat and fish. The food samples were inoculated with the virus, which remained infectious on fish, chicken, and pork, at 4 °C, −20 °C, and −80 °C, up to three weeks of storage [33]. Dai et al. (2020) demonstrated that SARS-CoV-2 can remain infectious on salmon at 25 °C for 2 days, and at 4 °C for 8 days, confirming that this virus is much more persistent at low temperatures [34]. Dhakal et al. (2021) and Jia et al. (2022), of the same research group, investigated the survival of SARS-CoV-2 on several food items [35,36]. In the first study, the virus was inoculated on food surfaces at 4 °C and observed for 24 h. It easily survived on chicken, salmon, shrimp, and spinach, while for apples and mushrooms the recovery was significantly lower, and the virus became undetectable in mushrooms at the end of the day [35], demonstrating that the virus prefers surfaces of animal tissues, in comparison to vegetal tissues [32,37]. In the second study, the survival was assessed on inoculated refrigerated and ready-to-eat deli items, fresh produce, meat, and seafood, for a longer period. In foods with higher protein, fat, and moisture content, such as meat and deli items, SARS-CoV-2 remained infectious for up to 21 days. Instead, in processed meat products, such as salami, and some vegetables, survival was strongly reduced. It is also interesting to note how the virus remained infectious in rare or medium-cooked beef, but not in well-cooked beef, demonstrating the importance of adequate cooking [36,38].

Regarding the persistence of other coronaviruses on fresh food, Mullis et al. (2012) studied the persistence of bovine coronavirus on romaine lettuce, stating that this specific virus, unlike SARS-CoV-2, became more stable with increasing temperature and increasing relative humidity and that it was detectable on the fresh vegetable for at least 14 days [39]. Van Doremalen et al. (2014) reported that MERS-CoV survived in dromedary camel milk at 4 °C for 72 h, while the infectivity was lost at 22 °C after 48 h [40]. Instead, Yépiz-Gómez et al. (2013) and Blondin-Brosseau et al. (2021) studied the persistence of HCoV-229E, a surrogate for the pathogenic coronavirus, with similar behavior [41,42]. The tested products were lettuce for the first study, and apples, tomatoes, and cucumbers for the second study. On lettuce, the virus disappeared at 4 °C within 4 days [41], while on apples and tomatoes infectivity was not detected after 24 h, and on cucumbers, it lasted until the third day [42]. This can be explained by the fact that coronaviruses are more stable at neutral than acidic or alkaline pH [43]. In fact, cucumber and lettuce have a pH of around 5.8, while apples and tomatoes have a pH of 3.9 and 4.2, respectively [42]. The observation for apples is in agreement with the aforementioned data on SARS-CoV-2 [35]. Therefore, it can be said that MERS-CoV and HCov-229E are less resistant to low temperatures than SARS-CoV-2, even if the tested foods were different [6]. These studies underline the persistence of coronaviruses on fresh products that cannot be treated with heat, thus allowing the survival of these viruses under refrigeration for a long time [6]. More recently, Bailey et al. (2022) conducted a persistence study of a lipid-enveloped RNA bacteriophage, phi 6, and of two animal coronaviruses, murine hepatitis virus (MHV) and transmissible gastroenteritis virus (TGEV), as SARS-CoV-2 surrogates, in meat and fish tissues [44]. The investigation demonstrated survival at both 4 °C and −20 °C for the entire period (30 days), with some differences of virus reduction among the four matrices (salmon, beef, pork, and chicken), and a general greater reduction of the virus in refrigerated than frozen conditions [44]. The results agree with the previous study on SARS-CoV-2 by Fisher et al. (2020), showing greater cold resistance of the tested viruses than MERS-CoV and Hcov-229E [33,44].

Assuming that the product is contaminated during the harvest by a sick operator, the risk of infection for the consumer is low because the virus is likely to lose its virulence, but if the contamination occurs at the end of the food processing chain, for example in the restaurant before the meal is served, the risk could be more real [42]. However, it must be considered that the SARS-CoV-2 inoculum used in most of the reported studies was more concentrated than what could be in real-world conditions, and so not fully representative of potential natural contamination [42]. In fact, in some studies, the inoculated virus was around 10^4^ [36] and 10^6^ [35] PFU (Plaque forming units) on 1 g of food or on each droplet spread over the food surface (usually 10 or 20 µL). Yu et al. (2020) reported that COVID-19 patients can release approximately 1.7 × 10^4^ copies of SARS-CoV-2 with a normal cough, which can potentially reach a food surface [36,45]. However, it would be necessary that the cough occurs directly and only on the surface in question, and that all the copies remain viable until consumption of the food, which is exposed to different environmental conditions during processing and handling for varying periods of time. Given these considerations, it is difficult that the ideal situations described in the studies can be encountered in real life. In fact, whatever the modality, there are no reports of contamination through fresh produce, so this route of infection is highly unlikely. Since there is no evidence of foodborne transmission, the typical epidemiological investigation of foodborne infections cannot be applied in this case; for example, patients with SARS-CoV-2 are not required to recall the food they have consumed [18,42].

In summary, SARS-CoV-2 prefers wet to dry surfaces and low to high temperatures, at which it resists better than other coronaviruses (up to three weeks in both refrigeration and freezing). It grows better in animal than vegetable tissues, especially if undercooked and less processed, and prefers neutral or alkaline pH.

### 3.2. Detection of SARS-CoV-2 in Foods

The reported data on the persistence of foods have raised hypotheses about the possible role of imported foods as a vehicle for the transmission of SARS-CoV-2 in places where unexplained outbreaks have spread or with apparent eradication [33,46,47]. In June 2020, the virus was detected on the surface of a cutting board for salmon, in Beijing Xinfadi wholesale food market (China). The discovery was made because a 52-year-old man was found positive for COVID-19 but had no contact with infected people. Sampling was made on the environments he had frequented, and two environmental samples from Xinfadi Market were found positive [47]. Therefore, China suspended the import of salmon from Europe and the import of many other food products from countries with the emergence of new outbreaks [33].

After this case, there have been numerous other reports concerning foods imported in China [37]. In most of these cases, the virus was on the packaging materials of frozen shrimps imported from Ecuador, but on one occasion it was also detected on the surface of a frozen chicken wing sample imported from Brazil, which became the first case of isolation of SARS-CoV-2 on a real food sample [30]. In September 2020, SARS-CoV-2 was detected on the outer package of frozen cod in Qingdao (China) [48], and in November 2020, it was found on the outer packaging of frozen pork imported from Germany [46].

The contamination of foods and their packaging by SARS-CoV-2 can occur in several ways during the “farm-to-table” lifecycle: in fact, during farming, processing, storage, transport, and retailing, foods come in contact with a large number of workers and with different environments that can be potentially contaminated [30]. These events can diffuse the idea that food transmission could represent a systematic risk for the spread of the virus. However, real data show that this is not demonstrated [49] and that it is unlikely that imported food or its packages, which have been transported and exposed to different temperatures and conditions for long periods, can carry an infectious virus [50]. In fact, this coronavirus can survive for certain periods on food surfaces and packaging but unlike bacteria, it cannot proliferate in food [38] and needs a host cell to multiply. Indeed, the detection of viral RNA on food does not mean infectivity and risk for the population [35,36].

In conclusion, reported data show evidence about findings of the virus on food surfaces, especially frozen, and of measures taken accordingly. However, as no cases of COVID-19 transmission through food consumption have been reported, it is not proved that suspending imports can contain the spread of COVID-19.

### 3.3. Persistence on Food Packaging and Work Surfaces

As already mentioned, SARS-CoV-2, like other types of coronaviruses, is stable at frozen and refrigeration temperatures, preferring low to high temperatures [33]. Regarding the type of surface, different materials with a wet surface could represent a substrate for survival. Considering a possible food transmission of SARS-CoV-2, not only food surfaces but also other materials must be considered, namely packaging materials, work surfaces, machinery, and work tools in food companies and in slaughterhouses. Packaging materials include plastic materials, especially polyethylene (PE), polypropylene (PP), polyethylene terephthalate (PET), polyvinyl chloride (PVC), and ethylene vinyl alcohol (EVOH), but also cardboard, molded paper pulp, and different metals [51]. In the food industry, most of the food contact surfaces and tools are made of stainless steel. According to van Doremalen et al. (2020), SARS-CoV-2, similarly to SARS-CoV-1, can remain infectious for up to 1 day on cardboard and up to 2–3 days on plastic and stainless-steel surfaces. On copper, the stability appears to be lower (4 h) [29]. However, in all the cases described by van Doremalen et al. (2020), the viral load gradually decreased over time. Comparable results were reported by Chin et al. (2020), with the longest survival being observed on smooth surfaces such as steel and plastic [52]. Riddell et al. (2020) demonstrated that inoculated virus, at optimal conditions (20 °C and 50% relative humidity), can remain viable after 28 days on non-porous glass, polymer, stainless steel, vinyl, and paper surfaces. On the contrary, at 40 °C, the virus was detected for less than 48 h for all the tested surfaces. At this temperature, the reduction of the viral load was greater than 4-log in all cases [53]. Compared to the study of van Doremalen et al. (2020), Riddell et al. (2020) used a more concentrated inoculum (at least 2-log higher), and this could explain the longer viability [29,53]. Kratzel et al. (2020) showed that on metal surfaces the viable virus was detected for several days up to 30 °C [54]. Liu et al. (2021) observed that the virus can persist for at least 60 days on the surface of cold-chain food packages. However, it was detected only 23 times out of 1360 swabs made on these surfaces (1.69%) [55].

It must be considered that all the tests were carried out In ideal laboratory conditions, and therefore in the real environment, with numerous additional variables, the persistence could be lower [56]. According to Lewis (2021), the contagion through fomites is almost nil, and despite this, too much importance is still being given to the disinfection of surfaces, while much more attention should be paid to the main infection route, which is air transmission [57]. In agreement with these findings, Sobolik et al. (2022) developed a quantitative microbial risk assessment model for frozen food packaging, from which it emerged that the decontamination of food packaging could be an excessively prudent and ineffective measure [56]. Conversely, continued exposure to disinfectants could lead to health risks [58]. On the other hand, the results of the study revealed the importance of the use of masks, hand washing, and vaccinations to reduce the risk of transmission. Furthermore, testing frozen foods for SARS-CoV-2 would not be convenient because of the costs, and the delays and inconveniences in distribution [56].

In summary, from the reported studies it emerges that SARS-CoV-2 shows similar persistence characteristics on food and surfaces such as packaging, machinery, and tools in contact with food (preferences for wet surfaces and low temperatures), and comparable survival times. Furthermore, it seems that the intensive disinfection of these surfaces is not a recommended practice, because contracting SARS-CoV-2 from food packaging or work surfaces is remote, and, as for food, there is no evidence of transmission through this route [59]. This position is shared by the CDC [60], which stated that the transmission of the coronavirus through contact with a contaminated surface is very rare, with less than one case in ten thousand.

### 3.4. Disinfection Methods

Despite these observations, to avoid even the slightest risk, food business operators should adopt adequate hygiene control measures throughout the production chain, including cleaning and disinfection of work surfaces and environments [61]. Enveloped viruses, including coronaviruses, are less resistant than naked viruses because they are easily inactivated by acids, detergents, disinfectants, drying, and heat treatments [27,62]. In particular, as their genetic material is coated by a lipid layer, coronaviruses are inactivated by substances that dissolve fat, such as surface-active agents and alcohol [63]. Several common disinfectants, included in the list provided by the Environmental Protection Agency (EPA) [64], have proved to be effective against SARS-CoV-2. For example, the virus is inactivated after a maximum of 5 min of exposure to common agents such as solutions based on 0.1% chlorine, 70% ethanol, 0.5% hydrogen peroxide, 0.05% chlorhexidine, and 0.1% benzalkonium chloride [52,65]. Alcohol-based disinfectants can inactivate enveloped viruses by 70–80% even within one minute of exposure [65]. Iodine-based agents and disinfectants with more than 30% ethanol demonstrated virucidal activity just within 30 s of contact [66,67]. The same observation was made for a more environmentally friendly chemical, N-decyl dimethyl ammonium chloride or bromide [68].

Moreover, coronaviruses are easily inactivated by heat treatments, and for this reason, at the early stages of the pandemic, the WHO recommended cooking animal products and handling with care raw milk, raw meat, and raw animal organs [8]. Pastorino et al. (2020) applied three different heat treatments to the virus, with temperatures ranging from 56 °C to 92 °C, and times between 15 and 60 min. Regardless of the treatment, a 4-log reduction was observed [69]. Chin et al. (2020) tested the stability of the virus at different temperatures (4 °C, 22 °C, 37 °C, 56 °C and 70 °C) on different surfaces, observing great stability at 4 °C, and a high sensitivity to heat [52]. Ultraviolet lights are also effective against SARS-CoV-2, but their use against the virus is not intended for the disinfection of industrial facilities, being less effective and more expensive than chlorine treatments [70]. In addition to food companies and processing plants, it is essential to implement these measures also in restaurants and canteens, where cutlery, dishes or glasses used by infected people could be a vehicle for infection. Normal washing and drying of dishes, in the dishwasher at around 60 °C, is considered sufficient to eliminate the risk of contamination. If the use of dishwater is not possible, hand washing should be performed with water temperatures not exceeding 50 °C, using an adequate amount of detergent [71].

In conclusion, SARS-CoV-2 is easily inactivated by traditional methods (common disinfectants, heating treatments, normal cooking, normal washing of dishes). Therefore, to avoid contamination, no extraordinary techniques are needed, and these common means can be used, as long as they are carried out correctly.

## 4. Impact of COVID-19 on the Food System

### 4.1. Impact on the Home Environment

The appearance of COVID-19 had a huge impact on the health of the world population but also on many other aspects related to life, such as the food system. After the WHO declaration of a global pandemic on 11 March 2020, the most immediate effect was the emptying of supermarket shelves. In fact, the lockdowns that many countries of the world had to face caused anxiety about not having enough food supplies [72]. While before the pandemic, at least for the middle social class, 50% of food was bought in supermarkets and 50% in food services, in the first months of the pandemic almost 100% moved to supermarkets [23]. In fact, in contrast to many other business sectors, the pandemic offered a great economic advantage for supermarkets [73]. Due to the fear of going out and contracting the virus, consumer visits to supermarkets became less frequent but longer in terms of time [23]. The most purchased foods were the commodities, such as bread, flour, eggs, milk and milk products, and shelf-stable foods such as canned products (tuna, beans, etc.), pasta, rice, and dried fruit [23,74]. In some cases, supermarkets placed product purchase limits for specific foods that were particularly demanded, such as flour and yeast in the European Union. The purchase of these primary products was linked to a growing trend of home cooking and baking, reported in many European countries [75,76,77,78], often explained as a response to inactivity and boredom [78]. Moreover, it was observed that, due to the fear of contamination by SARS-CoV-2, increased hygiene measures were implemented both in the domestic preparation of foods and in the management of food products bought at the supermarket, for example by disinfecting the packaging. In the study by Mucinhato et al. (2022), it was observed that Brazilian consumers improved their hygienic behavior in food handling because they were influenced by risk perception [77]. Faour-Klingbeil et al. (2021) reported that consumers in Lebanon, Jordan, and Tunisia changed their behaviors in favor of more sanitation and disinfection practices, including more people using detergents to wash fresh fruit and vegetables, such as soaps and non-food bleach [79]. This practice could be very dangerous, and in fact, the authors highlighted that information campaigns for consumers on the correct use of disinfectants for both hands and food are fundamental. The lack and need for correct communication about this topic were also stated by Finger et al. (2021), who observed that even Brazilian consumers often did not use chemicals correctly [80].

On balance, the COVID-19 pandemic showed a significant impact on the zoonoses statistics in the European Union in 2020, as demonstrated by the data gathered by EFSA and European Centre for Disease Prevention and Control [81]. In fact, according to this report, the pandemic might have caused a net decrease in both reported human cases and notification rates in all the European Union countries. Different factors might have played a role, such as the shutdown of travel, limitations in sporting and social activities, the closing of restaurants and bars, quarantine, and the whole set of mitigation measures. During summer 2020, France reported a higher number of cases of campylobacteriosis, which was explained as possibly due to less severe measures against COVID-19 [81].

Indeed, the changes in home cooking practices had effects on the diet quality, not necessarily perceived by most of the population, with variable consequences depending on the population subgroups. In their descriptive analyses, Sarda et al. (2022) found that individuals with financial hassles tended to cook less, whereas two out of five people reported improvements in their diet quality [78]. In general, the consumption of meals at home led to a greater focus on healthy foods, especially fruit, vegetables, legumes, and cereals also to strengthen the immune system against the coronavirus [61,82]. Mertens et al. (2022) carried out a survey in Belgium between July and October 2021 and found that the first motive for food choice was the sensory appeal, both before and during the pandemic, while the importance of healthiness grew during the pandemic, especially in urban areas. Furthermore, interest in the natural content of food also increased, mostly in younger respondents [83]. Caso et al. (2022) demonstrated that Italian consumers during the lockdown had a healthier diet and were more involved in preparing food at home, with a reduction in junk food intake. At the end of the lockdown, these habits were maintained discontinuously, with less attention to a healthy diet and food preparation, but still reducing the consumption of junk food [84]. According to Li et al. (2022), the perception of different types of risk (health, food safety, or financial risk) determined different behaviors in Chinese consumers [85]. For example, consumers who perceived higher health or food safety risks increased the healthiness of their diet, but with increased food waste compared to the situation before the lockdown. Furthermore, consumers who perceived a higher food security risk showed a greater purchase of sustainable products, unlike consumers who perceived a higher financial risk [85].

On the other side, in some countries, namely Denmark and Germany, the consumption of foods with a long shelf life increased [86]. In other cases, the feeling of depression and resignation that characterized the lockdown induced what is known as “emotional eating” which is a greater consumption of carbohydrates and sugars, which, together with reduced mobility and physical activity [82,87], could contribute to the onset of obesity and cardiovascular disease that can cause serious COVID-19 complications [88]. Emotional eating during lockdown was a global phenomenon, perceived also in the Middle East and in North Africa [89]. Comfort foods and ultra-processed products, such as snacks and sweets were particularly demanded during lockdowns [90]. In a survey carried out in Italy in April 2020, Di Renzo et al. (2020) observed that 48.6% of a large sample of respondents perceived a weight gain [75]. In another investigation carried out in Italy from April to May 2020, Izzo et al. (2021) found that 81.3% of responders reported an increase in frozen food consumption [91]. In the same year and in the same country, the Italian Institute of Frozen Foods reported a very significant increase in sales of frozen pizza and snacks (+15.6%), corresponding to 90,746 tons [92]. Another observed trend was the reduction in meat consumption [82], likely affected by consumers’ greater perception of the risks associated with eating meat, including the emergence of new viruses. Short-term reductions in meat intake were also observed in other zoonotic outbreaks, such as SARS and swine flu [93]. In addition, at the beginning of the pandemic, many restaurants and bars stopped serving rare steak and meat, as a precaution measure against foodborne pathogens. Consequently, many meat companies ceased their production during the COVID-19 pandemic [94].

### 4.2. Impact on Commercial Activities

In addition to an increase in home meal preparation, another clear effect of the pandemic was the increase in online food shopping [95]. Home delivery and takeaway were seen as alternatives to eating out: fast, convenient, easy, and without danger of contracting the virus [96]. Numerous dedicated apps were developed to help consumers assess and choose food products and restaurants, while other apps were useful to estimate the queuing time out of the shops. The spread of these platforms represented a new source of employment born with the pandemic. On the other hand, the classic jobs related to catering, such as that of cooks, waiters, and other workers of bars and restaurants, saw a decrease, and in many cases these people lost their jobs. Delivery and takeaway often represented the only job opportunity for restaurants [23,96]. Inevitably, the closure of many commercial activities caused a great economic damage for the workers of the sector, and this is one of the implications of COVID-19 on the increase in poverty of some categories of Western society. Despite the current improvement in the general situation also due to vaccinations, which led to the resuming of many restaurants and bars, the recovery has not yet reached the pre-pandemic levels, and many people are still in financial trouble [96].

While many workers lost their jobs during the pandemic, there was also a shortage of manpower for certain types of work [97,98]. For example, multinational food companies had problems in meeting the demand due to the lack of workers, often migrants, who returned to their homelands [72,99]. Therefore, labor shortage was another big issue raised by the COVID-19 crisis, in the livestock, agriculture, and industry sectors [61]. Even after the first wave of COVID-19, in the following waves throughout 2021, supply side disruptions became more significant than demand-side disruptions [100]. Supply side disruptions, due to labor shortage, transport difficulties, or closure of borders, are still evident in 2022, in particular for the food items that need to be transported from distant lands [98].

### 4.3. Impact on Food Access

The pandemic also worsened the condition of the poorer populations. The disruption of national and international food supply chains, caused by travel restrictions due to the risks associated with the infection, led to food insecurity in many low-income countries but also in wealthy nations [73,101,102]. In fact, the closure of borders caused a lack of food resources in countries that depend on imports [100,103]. Moreover, the shortage of staple food caused an increase in prices, contributing to general poverty [97,99]. It is estimated that between 83 and 123 million people, including between 38 and 80 million people from countries that rely on imports, suffered from the food shortage brought about by the pandemic [104]. Moreover, food insecurity negatively affected health, leading to a high incidence of chronic diseases and infectious diseases, especially in older adults [105], and consequently is associated with a higher COVID-19 rate [106].

Many governments aided the population suffering from food insecurity, in the form of cash or food. In some African countries, safety nets in the form of cash significantly reduced food insecurity, while food assistance was less effective in this direction. This was because cash assistance allowed us to consider many aspects, such as the different needs in variable diets and nutritional intake [101]. Furthermore, cash transfers are generally cheaper, because they are easier to manage [107]. The same study [101] evidenced how, in African countries, the category most affected by food insecurity was that of female-headed households, and the poorest and least educated people in society.

Therefore, the pandemic questioned the concept of the resilience of the food system. This term was defined by Tendall et al. (2015) as the ability of a food system to ensure that everyone has adequate and sufficient access to food, even in emergency or crisis situations [108]. While before COVID-19 resilience was not seen as a problem, as the main issues were about healthiness and safety of food, with the pandemic this concept, which should be taken for granted, was questioned [23,101,109]. 

### 4.4. Impact on Food Loss and Food Waste

Another serious consequence of the pandemic was the remarkable food loss during production and processing stages, and food waste at the end of the food chain, due to the closure of commercial food businesses. Perishable foods, such as meat, fish, and vegetables, were the most damaged and lost [103]. In many cases, the supplies stored in bars and restaurants were damaged because they were impossible to use. This resulted in great economic losses, as well as severe damage to sustainability. To minimize the losses, temporary regulations allowed and encouraged the freezing of leftover food, especially meat [110]. Moreover, because of the strict rules in the food supply chain, food designated for food service could not be repackaged for retail sale [111].

However, food loss and food waste were particularly observed on the transport front, in which the maintenance of the cold chain was the crucial point [37,112]. This can be very difficult during all the stages, from production to trade, when temperature fluctuations can easily occur, causing food waste also in normal situations [113]. These problems were exacerbated by the pandemic since transport required higher costs and longer times (e.g., due to health screening of workers), and furthermore, the processes were slowed down by the shortage of workers. One example is given by the difficulties in cargo handling of goods traded by sea, but inconveniences were also widely observed in air, land, or railway transportation [112]. In these situations, and especially while loading or unloading goods, food products were exposed to unsuitable environmental conditions for a long time, with temperature increasing up to 10 °C [114], and thus spoiled. Moreover, the higher temperatures could increase viral activity on food surfaces and packaging, favoring SARS-CoV-2 transmission. For example, the detection of the virus on the outer package of frozen cod in Qingdao (China) [48], could be linked to improper maintenance of the cold chain during transport.

Staple crops were a food category largely lost because of the pandemic, especially due to the shortage of manpower and work stoppage [99]. Moreover, the waste of staple crops resulted from the lower demand from the catering businesses, due to their closure. Therefore, many businesses, especially smallholder farmers, lost their jobs [99]. Finally, the low demand for meat and dairy products determined the waste of these food categories as well. Indeed, in some cases, the farmers were forced to cull the animals because they could not find any plant to sell them [61].

Therefore, on the one hand, the demand for both staple and processed products increased with the pandemic, resulting in a shortage in supermarkets, on the other hand, the closure of bars, restaurants, hotels, and workplaces and the reduction of food trade had negative repercussions on the agri-food sector, with food waste all over the world. This contradiction derives from the fact that many countries have two distinct supply chains, one for grocery stores and one for food service. Thus, the emptying of the shelves due to the high demand for goods did not correspond to a real lack of that product but to the fact that that product was lacking in the form suitable for sale at the supermarkets [74]. However, in some cases, the passage of goods between the two distribution channels took place. This is the case of the Canadian fruit and vegetable market, which was forced to move all supplies from food service to retail. This resulted in numerous logistical difficulties, but the supply chain was able to remain robust and resilient [115,116]. On the other hand, the cattle/beef sector in Canada was already accustomed to emergency situations, because the system had already faced one in 2003, during the Bovine Spongiform Encephalopathy (BSE) epidemic. In fact, even in that occurrence, there was the destruction of the supply chain, and the measures put in place have been resumed in the current pandemic [117,118].

## 5. Outbreaks of COVID-19 Linked to Food Production Systems

### 5.1. Favorable Factors

The SARS-CoV-2 pandemic has touched all social classes and labor sectors, but food industry is among the most severely affected by outbreaks. This is due to many factors, including that in this sector it is difficult to work from home. The most involved industries were slaughterhouses and meat and seafood processing facilities. These environments present ideal conditions for the transmission of the virus, such as overcrowding, poor ventilation, and low temperatures [33,119,120]. In fact, SARS-CoV-2 can remain infectious at refrigeration temperature and on metal surfaces, which are very common in these plants. In particular, in abattoirs, the killing line works at room temperature, but the subsequent process takes place under controlled temperature conditions, around 12 °C, which are ideal for the spread of SARS-CoV-2 [121]. Then, the meat is kept at 3–7 °C [33]. Furthermore, air humidity at low temperatures, necessary to keep the meat from drying out and therefore not to lose weight, could contribute to the transmission of the virus.

Moreover, a fundamental role seems to be attributable to the air conditioning systems used to cool and ventilate the processing plants, which would favor the circulation of bioaerosol in which the SARS-CoV-2 is suspended [122]. Lack of UV light also contributes to the persistence of the virus [123]. In addition, due to the noise in the environment, workers speak loudly, and this facilitates the transmission from person to person [33]. The fatigue associated with the hard work also causes heavy breathing, the greater release of droplets into the air, and therefore greater contagion [123]. Slaughterhouses and meat and poultry processing usually require manual work. Instead, in more automated plants, such as those of salmon processing, with filleting and cutting performed by machines, there is no overcrowding of workers, and transmission is less probable [33].

### 5.2. Prevention Measures

Many measures, regarding work personnel, food, utensils, and work surfaces, can be put in place to reduce the risk of contamination and the emergence of outbreaks in such suitable environments. Workers should take care of personal hygiene, with proper hand washing before and after the shift, and correct handling of work uniforms, but they should also avoid overcrowding and use face masks correctly [21,24]. However, due to the structure of these plants and the type of operations, especially in line, workers are often unable to maintain a safe distance. This is very difficult also while entering and exiting facilities. In addition, food products must be handled appropriately during processing or slaughter, and the disinfection of tools, machinery, surfaces, and work environments must be done with the right frequency and with specific disinfection products. To make this happen correctly, adequate training programs for workers should be put in place. In addition, sick workers should be excluded from work, and financial support should be provided, so that they are encouraged to report their sickness and stay at home, not to pose a risk to other workers [33].

Furthermore, the workers of slaughterhouses and meat processing plants are often migrants from poor countries, living in conditions of scarce hygiene and overcrowding. They are often transported to work by buses full of people, where the virus can easily circulate [120]. Therefore, they should be ensured with adequate and worthy accommodation and transportation, to reduce the risk of infection. Another consideration about the outbreaks in meat processing plants is that in some countries, for example in the United States, companies requested and obtained permission to increase the speed of the production line, to meet the need for products during the pandemic. This has favored overcrowding and disorder in the processing lines, resulting in an increase in the number of COVID-19 cases. Therefore, it is very important to find a balance between ensuring the correct supply chain and preserving public health [124].

Despite the importance of the safety measures described above, not all of these are perceived as essential by the food industries. Djekic et al. (2021) carried out a survey involving 16 countries and 825 companies from which it emerged that the measures considered most important for the containment of COVID-19 were staff awareness and hygiene, rather than precautionary measures dictated by the World Health Organization, such temperature checking of workers. The use of gloves and masks was not considered a salient measure, also because it was already implemented for various operations before the pandemic. Hygiene measures in food companies, already implemented before the pandemic, were consolidated, and made more restrictive. Furthermore, the survey revealed that the most dangerous moment for the infection of the virus is in retail, while food storage activities were considered the safest. In any case, food business operators stated that food safety was never questioned, and in fact, only a few companies needed to develop emergency plans [125].

### 5.3. Consequences of Food Production Outbreaks

In general, the biggest number of outbreaks were seen in the largest meat plants, when the production line was faster, with pork, beef, and poultry processing plants showing the highest correlation with COVID-19 cases [124]. These outbreaks were often followed by lockdowns and periodic shutdowns of the plants. The United States was particularly hit by this phenomenon, with the closure of the first poultry farm in Louisiana, on 27 March 2020. Then, there was the closure of several other cattle, poultry, and pig plants [74,126]. Especially in the pig and poultry industries, this phenomenon caused a reduction in processing and slaughtering capacity, with overcrowding of animals on farms. Therefore, other slaughterhouses had to be found, often far away, but this was not always possible, and many animals had to be culled. As a further problem, there was the disposal of the carcasses, associated with potential biosecurity risks, and danger to the environment [74]. Closely related is the ethical aspect and the animal welfare: the hygienic conditions and the overcrowding to which animals were subjected, determined suffering and stress for them. In addition, in some cases the slaughterhouses, especially those for chicken, sped up the line to increase yield and save time, thus causing incomplete or incorrect stunning, with consequent suffering of the animal and reduction in the quality of the meat [74].

The weaknesses and limitations of the food system, for which these activities affected by the outbreaks had such serious consequences, were, first, the scarce insurance for workers, especially foreigners, who often live far from their homes without adequate accommodation, in conditions of overcrowding, thus favoring contagion. In addition, the fear of losing their jobs urged them to work even in case of illness, favoring the spread of the virus. Another weakness of the food system was the lack of adequate training for workers, which would be useful to better understand the risk associated with the pandemic, as well as the preventive actions to be implemented at each stage. Another serious mistake made by the companies was the increase in productivity, supporting the spread of the virus, also due to a greater release of droplets by tired workers, subjected to fast rhythms. To cope with these serious issues, it is necessary to increase the tolerance and resilience of the system in the face of the next pandemic, being able to change and adapt to be ready even in emergency situations. For example, food companies should invest in automation, which would reduce manual labor, crowding, and the risk of contagion, avoiding stressful shifts for workers. Furthermore, workers should be trained in good practices to follow according to the situation, and foreign workers should be ensured decent living conditions.

Examples of large clusters among workers, which often caused the closure of the plant, were in slaughterhouses, meat processing plants, meat-packing factories, and abattoirs in the United States, Canada, Brazil, China, Ghana, Australia, United Kingdom, Portugal, Ireland, and Germany [119,123,127,128]. Thousands of COVID-19 cases were reported in slaughterhouses in Germany, and therefore the German Federal Institute for Risk Assessment (BfR) issued an opinion to ensure that the meat does not pose a COVID-19 risk for consumers [129]. However, plants were closed due to the shortage of workers infected with COVID-19, and not the transmission of the virus from raw meat [130]. For this reason, the FDA (2020) stated that food must not be recalled from the market [18].

## 6. Guidelines and Recommendations for the Food Sector

Governments of countries around the world, together with national or international bodies, have introduced various containment measures to react to the pandemic, also in the food sector. These have been applied more or less restrictively from the beginning of the pandemic, depending on the severity of the epidemiological situation, and were mandatory, especially in the first months of lockdown. In the current period, many restrictions have decreased in most of the world, but this is not the case in some countries, such as China, which continue to adopt them [131]. In addition, many people in several countries continue to use face coverings and maintain social distancing, even if these measures are no longer mandatory, for prevention. The following guidelines and recommendations were the main ones applied during most of the pandemic.

The use of face masks was generally mandatory, in the workplace, at least in indoor ones, as well as the maintenance of a safety distance between one person and another. This distance is at least 1 m (3 feet) according to the WHO [20,132], but increases in some countries, such as in the United States where it is 6 feet (around 1.8 m) according to the CDC [133]. Indeed, it is estimated that respiratory droplets can travel a distance of about 1 m when breathing, 1.5 m when talking, and 2 m when coughing [134]. These provisions were applied to workers of food companies but also to customers of restaurants, bars, pubs, and cafes. For catering activities, the seating capacity was limited to ensure the minimum safety distance, both for indoor and outdoor seats [135]. Therefore, the arrangement of the tables was modified to maintain this requirement. Commercial establishments, both restaurants, and supermarkets, posted near the entrance door the indication of the maximum number of people that the place can hold at the same time. Queues were avoided by shifting customer booking times in restaurants. Markings on the floor were recommended to suggest maintaining a safe distance where there was much more users’ concentration, such as in correspondence of the entrance or at the service counters of supermarkets [135].

Shops and restaurants were equipped with products for hand sanitization in multiple positions, especially at the entrance, where users were encouraged to use them through visible signs. In addition, devices for measuring the body temperature of workers or customers at the entrance were introduced [50]. Customers had to wear a mask during all activities other than meals at the table, for example during payment and use of toilets. It was advisable to use alternative presentation formats of the menu compared to the traditional ones, for example, available via apps and websites, or menus of the day printed on disposable sheets. There was no valid evidence to suggest switching to disposable tableware. Therefore, it is advisable to continue with the use of traditional dishes, glasses, and cutlery, ensuring correct washing, cleaning, and sanitation, when possible, by means of a dishwasher. Buffets and self-service activities were allowed or not, depending on the country and the severity of the COVID-19 situation but, when permitted, distancing between customers and hand sanitation before the use had always been ensured. Electronic payment, instead of cash, were recommended, and, where necessary, insertion of separatory barriers at the cash desks were provided [50,96]. In closed rooms, it was recommended to ensure periodic ventilation, if possible natural. For ventilation, maintenance of artificial systems was necessary, especially by cleaning and replacing the filters. Anyway, in all steps it was recommended to strengthen hygiene measures, optimizing the management of cleaning and disinfection [136].

The implementation of these measures was essential not only to reduce the contagion but also to encourage a return of consumers to the bars and restaurants that reopened after the periods of closure. In fact, as demonstrated by Vandenhaute et al. (2022) with a survey carried out in Belgium, consumers who perceive that safety and hygiene measures are effectively applied are more encouraged to eat and drink out [137].

Apart from these preventive measures, other interventions by governments, such as school closures, travel restrictions, bans on gatherings, emergency investments in the healthcare sector, closures of restaurants and other commercial activities, and trade restrictions, were implemented in different countries at different times of the pandemic, to minimize the risk of contagion [138]. These were more severe in the first months of the pandemic, with a peak in April 2020, and mitigated in the following months, undergoing a variable trend during the multiple waves [101]. Some are still in place in 2022. These measures, particularly those on restrictions on business activities, outlined how people’s health was deemed more important than economic interests [139]. Despite this, the strong impacts on society determined bad consequences for many people, as previously described.

## 7. Lessons Learned for the Food System and Research Needs for the Future

The current pandemic has changed the lives of people all over the planet and has profoundly affected all activities in the food sector. Lessons from the past were often ignored. For example, although it was quite clear from the beginning that social distancing, face masks, and closure of places of aggregation were effective measures, many countries delayed the application of these measures, and the effect was the global dimension of the disease [140]. However, the outbreak of COVID-19 can be considered an opportunity to learn what we have missed, to be able to cope with future pandemics. A summary of the lessons learned from the pandemic, described below, is proposed in Table 1, with divisions based on different areas of interest regarding the food system.

One of the most important lessons learned is that crowding should be avoided in any context, since this virus and other viruses are transmitted mainly by air and in closed places [24]. Therefore, the density of people should be controlled both in factories and in shops, through the rationalization of activities both in production and distribution. In food companies, where possible, the automation of the lines should be envisaged, to guarantee distancing and less stressful shifts for workers. This modification of the production system could be particularly useful in slaughterhouses, which have been the scene of numerous SARS-CoV-2 outbreaks. On the other hand, in supermarkets, gatherings are created mainly at the entrance. To avoid it, the development of existing but not very widespread apps could be promoted to alert customers about the progress of the queue. Moreover, to prevent crowding near the counter, fresh products could be conveniently supplied already packaged, and this could also increase the confidence of consumers, who prefer to buy a product protected from external agents, including viruses.

Another lesson learned is that the organization of the spaces and most of all air conditioning can play a fundamental role in contagion, both in industries and restaurants. Specifically, air purification technologies need to be improved [96]. In this respect, a cross-sectional study of one of the greatest outbreak events of COVID-19 in a meat plant in Germany [127] demonstrated that breaking the rules of social distancing but also climate conditions and low outdoor air flow played a significant role. The importance of the direction of the airflow was highlighted in the first months of the pandemic, when an outbreak in a restaurant in Guangzhou, China, was linked to air-conditioned ventilation [141]. On that occasion, the direction of the airflow in which the droplets were suspended was the same as the virus transmission [141]. Previously, air distribution played an important role also in SARS transmission [142]. Nevertheless, the measures applied in the world during the COVID-19 pandemic focused more on social distancing than on confined space air quality. Further studies are needed to better investigate the role of airflow in the outbreaks of COVID-19 in restaurants and in the food industry.

An important lesson learned during the lockdowns is the value of online commerce, which could be promoted among small traders and farmers to sell their products without the need for contacts, and therefore in safe conditions, to enhance their productivity, avoid food waste, and increase their profits because of the reduction in the number of intermediaries [23,99,143]. Moreover, delivery and takeaway practices represent opportunities for restaurants to continue working safely and efficiently [96]. An example is reported in the study by Paul et al. (2021c), which evaluated a model for the recovery of online businesses, to help traders to make the best choices to implement an optimal recovery plan, maximizing profit [144].

Another important issue raised from the pandemic is the maintenance of the cold chain during food transport, to minimize both food losses and food waste, as well as the survival of SARS-CoV-2 on food surfaces. In this regard, correct practices established by international standards (e.g., ISO 9001, ISO 22000, FSSC 22000, HACCP, but also ISO 26,000 that represents an international standard for Corporate Social Responsibility) and specialized agencies (e.g., EFSA for European Union and FDA for the United States) should be carefully followed during food transport, but also in all other phases of food handling [112]. Additionally, for greater efficiency in accomplishing this purpose, Statistical Process Control (SPC) tools are increasingly recommended for food quality and food safety control, including monitoring transport temperatures. These methods are supported by new technologies, such as the Internet of Things (IoT), a complex system that collects data from a multitude of sensors, such as Radio Frequency Identification (RFID) sensors, allowing not only real-time signaling of thermal abuses, but also to investigate the causes and prevent future inconvenience [112]. Future research in this innovative and promising field is desirable, to implement a system of traceability and maintenance of the cold chain during food transport. Moreover, predictive models, such as the one proposed by Shahed et al. (2021), could be useful to limit disruptions along the supply chain at the supplier, manufacturer, and retailer levels [145].

In addition, this pandemic also demonstrated the importance of proper management and reorganization of wet markets where live wild animals are traded. Although the origin of SARS-CoV-2 is still uncertain, it is likely that it is linked to a wet market [13,15], and other new viruses could develop in these environments. In fact, the animals are kept in cages that are placed in stalls very close to each other, and they can be slaughtered even instantly upon customer request. Considering that these markets provide millions of people with fresh and low-priced food, they need to be regulated, as suggested by WHO, by hygiene practices and space management [23,146]. The same considerations should be made for intensive farming, where the cattle are often crowded and in stressful conditions, with a greater risk of emerging zoonotic diseases [147].

Another lesson that should be learned for future emergencies is to avoid supply side disruptions, observed in different areas of food production, such as in agriculture, livestock, and aquaculture supply chains, as opposed to demand pressure [97]. This was caused by a lack of manpower, transport difficulties, food losses, and the closure of borders. The lack of manpower happened because workers, often foreigners, in many cases returned to their homelands after the pandemic. To avoid this phenomenon, they should be provided with protection policies that allow them to feel safe even away from home, for example with incentives, suitable accommodation, promotion of vaccines, coverage of healthcare costs (e.g., coronavirus testing), reduction of bureaucracy for their work and permanence permits, in order to encourage them not to leave their workplaces in case of future emergency situations [97,139]. On the other hand, protection of trade and supply of goods, especially staple food, and minimization of food waste should be performed through special political agreements between countries. Reduction of food losses and waste should also be reached with the control of the correct conditions during the transport of food, as mentioned above.

Directly linked to the previous issue, there is a need to ensure global food security. This point was questioned during the pandemic, as the poorest populations did not have normal access to food, due to its shortage and price speculation [97]. This was often followed by malnutrition and worsening of general health conditions, with a greater likelihood of infection by SARS-CoV-2. Food protection guarantees and insurance policies for these social groups, in some cases, already implemented [101], should be performed systematically in the event of future pandemics. For example, actions to regulate prices of essential products and aid in the form of cash or food should be provided [101,138,139]. Additionally, industries should implement organizational plans to ensure basic food products. One strategy could be the use of alternative ingredients, possibly locally available and therefore easier to find [23].

Directly connected to ensuring global food security, there is the creation of a sustainable food system able to guarantee healthy food to all social groups, by realizing the “Farm to Fork Strategy”, which is part of the European Green Deal (EGD) implemented by the European Union. This program foresees that every actor of the food chain (from agriculture, livestock, and aquaculture to the consumer) has a specific role, but is connected to the others, in building a food system that ensures global food safety [148]. The program pays particular attention to the sustainability of the system, particularly in primary production, promoting a circular economy [97,139]. This includes the reorganization of intensive farming in a more sustainable way, but also, in the agricultural field, greater attention to green and innovative technologies, as proposed by Rowan and Galanakis (2020) [149], the reduction of the use of mineral fertilizers and pesticides, and the promotion of local crops, with high nutritional value [139]. Aquaculture systems are also involved, for example implementing multitrophic systems, which integrate fish, mollusks, crustacean, and seaweed farming into an integrated ecologically neutral system [139,150].

Therefore, COVID-19 should give rise to a new food system, which allows the sustenance of the world population by guaranteeing food safety and reducing waste [146,147]. This means ensuring the resilience of the food system. This concept is of crucial importance because it represents the capacity to change as a response to the emergency, but also to continue to develop. An example was the 1918 flu, which led to the modernization of health systems. Similarly, it is essential that all activities in the food sector are reorganized in production, post-harvesting handling, processing, supply chain, marketing, purchase, and consumption, according to strategies that allow maintaining resilient and sustainable practices [109,130,138,149,151,152]. In this context, the previously described “Farm to Fork Strategy”, could assume a crucial role [97,139]. For this purpose, a collaboration between the food system and the public health system is also recommended, as provided by the One Health approach [153]. According to it, health policies recognize the food system as the basis of the health system, while agri-food policies recognize public health as the main goal to consider in their decision-making processes [138].

Another outcome highlighted by the pandemic is the importance of correct information. On balance, the virus–food relationship is still a little debated, and more attention should be paid to this topic through the contrast of the fake news. In fact, although the transmission of SARS-CoV-2 through food is considered unlikely, this could happen for other viruses. The appearance of other zoonoses or other diseases in humans is probable in the future and could lead to future pandemics, favored by the increasing world population density, globalization, and therefore ease of movement from one area to another of the planet, but also by intensive exploitation of natural sources, environmental catastrophes, and climate change [154]. The current emergency should therefore teach people to cohabit with the virus, providing proper information, but also to take actions to prevent its further appearance. Overall, when foods are suspected to play a role as a vehicle of viruses, good hygiene measures must be implemented correctly, and this implies food hygiene knowledge among consumers [8,120]. Competent authorities, as well as specific organizations, such as FAO (Food and Agriculture Organization of the United Nations), should educate citizens on these issues, as well as on the role that food could play in carrying viruses. For this educational purpose, information programs should be offered to people through communication channels such as social media, but also more specific programs to be supplied to food companies, which in turn should provide it to their workers.

Although SARS-CoV-2 cannot be defined as a foodborne virus and is not subjected to a typical epidemiological investigation of foodborne infections (for example there is no recall of food consumed by COVID-19 patients) [18,42], epidemiological surveillance measures could be put in place to check for any future evidence of the connection of SARS-CoV-2 with food, as already performed for other viruses [155,156], to further understand the link between this virus and the food system. In this regard, quick response teams could be exploited, with operators at each point of the food chain, and case investigation methods would be useful for collecting data for epidemiological studies.

Finally, it should be considered that the limitations of this study are given by the fact that the scientific evidence mainly derives from cases of SARS-CoV-2 detected in the food system, and less from studies on food models, which, even when completed, are not highly standardizable, differently from what happens with traditional foodborne microorganisms. For example, several SARS-CoV-2 persistence studies have been performed on foods, whose composition can be highly variable not only in different manufacturing environments but also during the shelf life. More reliable information can be obtained in the studies on fomites, e.g., work surfaces of food companies. Therefore, further studies should be performed in this field, to obtain information that can be considered representative of the different conditions encountered in the food system. Meanwhile, this review gives an overview of the experience gathered so far but also of the work that still needs to be done.

## 8. Conclusions

This review demonstrates that there are no proven cases of transmission of SARS-CoV-2 through food, packaging, work surfaces, or work tools and that a foodborne transmission is highly improbable, even if the virus can persist also for a long time on food surfaces. In fact, it is possible that the virus may have been accidentally ingested if present on the surface of a food, but there is no evidence that this route has caused disease. Moreover, this research shows how the COVID-19 pandemic has impacted the food system, in detail consumer habits (increased demand for more natural and healthier foods, decreased consumption of meat, especially raw meat, greater attention to hygiene practices, etc.), commercial activities (closures and job losses, shortage of manpower, more interest in online sales), food access (limited for poorest populations), and food loss and food waste (increased especially because of closure of activities and failure in maintenance of the cold chain during transport of goods). Furthermore, this study demonstrates how some food systems, in particular meat processing facilities, are ideal environments for the transmission of the virus, and therefore linked to numerous outbreaks. These findings are of crucial importance because the implementation of adequate preventive measures (hygiene in all the processing phases, correct use of personal protective equipment, maintenance of interpersonal distance, and adequate training programs) could avoid the development of future outbreaks. In addition, reporting the major guidelines applied during the pandemic, and the lesson learned (such as the importance of hygiene practices both in the domestic and work environment, the value of alternative commercial channels, such as online sales, the reorganization of food activities, in particular, those selling live animals), the review highlights how much important is that the food system remains resilient during this pandemic and in view of future ones, putting in place strategies that make it possible to guarantee the quality of life of all peoples even in emergency situations and to ensure activities and trade. The experience gained during the pandemic will probably have long-term or permanent effects on the food system, by accelerating automation and digitalization, in line with the lean manufacturing principles, as well as by developing new solutions to improve environmental conditions in the food industry and work organization in restaurants and supermarkets. The lessons learned from COVID-19 will be fundamental to fighting against future pandemics and preventing the onset of new zoonoses within the food system. Moreover, future studies are recommended to deepen the knowledge on the relationship between viruses potentially responsible for pandemics and the food system, as well as the factors which contribute to determining the occurrence of new zoonoses, and the role of environmental factors in the spread of the virus.

## Figures and Tables

**Table 1 foods-11-02816-t001:** Lessons learned from the pandemic in various areas of interest in the food system.

Areas of Interest	Lessons Learned
Primary and secondary sectors (production and industry)	Importance of good hygiene practices Rationalization of activities to avoid overcrowding Importance of personal protective equipment Increased automation and digitalization Value of online commerce Improvement of air conditioning purification technologies Maintenance of the cold chain (also using new technologies) Proper management and reorganization of intensive farming Protection policies to counteract labor shortage Plans to ensure basic food products Greater attention to green and innovative technologies Ability to reorganize according to the needs (resilience) Importance of correct information Educational programs Implementation of surveillance measures Study of the relationship between the food system and potential new zoonotic agents
Service sector (restaurants and supermarkets)	Importance of good hygiene practices Rationalization of activities to avoid over-crowding New tools to provide information (e.g., menu QR codes) Importance of personal protective equipment Increased automation and digitalization Improvement of air conditioning purification technologies Importance of the natural air circulation Apps for queues monitoring Value of online commerce Maintenance of the cold chain (also using new technologies) Policies for protection of trade and supply of goods Ability to reorganize according to the needs (resilience) Educational programs Implementation of surveillance measures
Home environment	Importance of good hygiene practices Improvement of air conditioning purification technologies Importance of the natural air circulation Maintenance of the cold chain Importance of correct information Educational programs
Society	Importance of good hygiene practices Rationalization of activities to avoid over-crowding Importance of face coverings Plans to ensure basic food products Ability to reorganize according to the needs (resilience) Importance of correct information Educational programs Implementation of surveillance measures Study of the relationship between the food system and potential new zoonotic agents Improvement of the organization of the national health system
Food trade	Importance of good hygiene practices Importance of personal protective equipment Increased automation and digitalization Value of online commerce Maintenance of the cold chain (also using new technologies) Protection policies to counteract labor shortage Proper management and reorganization of wet markets Policies for protection of trade and supply of goods Plans to ensure basic food products Greater attention to green and innovative technologies Ability to reorganize according to the needs (resilience) Importance of correct information Educational programs Implementation of surveillance measures Study of the relationship between the food system and potential new zoonotic agents

## Data Availability

Not applicable.

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
