# Peer review of "The Multifaceted Relationship between the COVID-19 Pandemic and the Food System"

_foods, 2022, doi:10.3390/foods11182816_

Round 1

Reviewer 1 Report

The manuscript is of a review nature, which allowed the authors to structure the content freely. It is correct. The article consists of an abstract and 6 chapters:  1.Introduction, 2. Pertinence and detection... (on food, in foods), 3. Impact of COVID-19 on the food system (in house, on commercial activities, impact on food access and impact on food waste), 4. Outbreaks of COVID-19 linked to food production systems (favourable factors, prevention measures, Consequences of food production outbreaks), 5. Guidelines and recommendations and finally 6. What to do about future. The layout of the paper is based on a deep study of the current literature (more than 150 items) and previous observations and experiences.

Comments

I believe, however, that the content lacks more extensive mention of: 1) the role of transport in combination with temperature monitoring of food during transport and in trade, 2) the necessity of SPC tools in a food quality control system supported by new technologies (RFID sensors and IoT). Marginal treatment is also given to the problems of food losses as opposed to food waste.  This is a certain shortcoming in relation to the circular economy as a desirable model for shaping the food system in the 21st century.

Another shortcoming of the work is the lack of figures, graphs and tables with data to make the argument easier to understand and remember. In addition, I believe that the paper should mention the role of systems such as ISO 9000, ISO 26000 and HACCP. As the article is extensive I suggest including at least a figure or table of its results. 

The authors demonstrated that during the pandemic, food purchasing patterns and diets changed and that there was an increased demand towards more natural, healthy foods with a short shelf life.  They further showed that consumption of meat (especially raw meat) and direct consumption of other products, for example, decreased. At the same time, online sales have increased. They also point out the lack of workers in food processing.

Based on the available literature, they rightly call for more social distancing and increased attention to the role of temperature in restaurants and forced air circulation in the food industry during a pandemic attack. They also suggest undertaking more research on the effects of temperature on food storage and handling. It is also important and right to point out that virus-food interactions require further basic and applied research, and perhaps a modernisation of the health system. The suggestion to control the number of customers in retail outlets (especially food outlets) is also correct. Such systems are already available (e.g. Hitachi. Controlling COVID-19 Retail Traffic with Automated People Co. Available online: https://global.hitachi-solutions.com/blog/automated-people-counters).

Reviewer 2 Report

Refer to recent paper on epicenter of outbreak:  https://www.science.org/doi/10.1126/science.abp8715

A methods section is not provided that provides details on how the review was conducted so difficult to evaluate scientific merit.

Section 5: Is this a review of current guidelines?

Conclusion: Unsure of the basis for recommending: ,"further studies are needed to deepen the knowledge on the relationship between the virus and the food system, acquiring more robust data regarding the fact that foods do not represent a significant route of contagion."  More evidence is presented on the the role of meat processing facilities in facilitating infections among workers so this likely deserves more focus in the conclusion.

Reviewer 3 Report

The article submitted for review is a study of the current literature on the occurrence of SARS-CoV-2 in food. The material is presented in 7 chapters, the first and the last of which are the introduction and summary of the work. Chapter 2 lists the literature reports confirming the presence and detection of the virus in food, the survival of surfaces and packaging materials that come into contact with food, and the effectiveness of disinfectants. Chapter 3 deals with the impact of the Covid 19 pandemic on the household environment, the type of dietary changes due to the availability of food, ways of shopping, and the impact of the pandemic on food waste. In Chapter 4, the authors raised important issues related to the factors contributing to the transmission of the virus as well as the factors preventing it and the effects of infections in food establishments. Chapters 5 and 6 respectively describe the recommendations for action in the food sector to limit virus transmission in the conditions of production facilities and a summary and reflection on the mistakes made in ensuring food safety during the Covid-19 pandemic.

Overall, I believe the article is well written and detailed, and the authors have sufficiently substantiated the importance of the study. The authors note that food safety issues include the emergence of a new threat such as the Covid-19 virus as well as a pandemic phenomenon across the food chain in countries and around the world.

Reviewer 4 Report

The paper is a comprehensive review of the possible relationships between the coronavirus outbreak and various animals, types of food, food packages, and other components of food systems. Being a non-original study of the secondary sources, this manuscript could still go under the review category. However, I would like to recommend the author to revise the abstract by outlining the major findings and future research directions or potential outcomes of the study

Reviewer 5 Report

Comment to authors

Abstract

The abstract can be written more briefly (especially the four first lines) and the important findings of this review should be stated in the result section.

Furthermore, the conclusion seems to be too general. The conclusion should be completely based on the results obtained.

Introduction

The first paragraph of this part is not necessary and should be summarized, as this review focus on food topic. These days everyone knows the history and type of SARS viruses.

Line 44: Please use “human” instead of “men”.

According to the sentences in lines 59-63, and based on the evaluation of food agencies, what is the need to investigate the issue that has been fully evaluated and the results announced? Please give an important issue here, exactly after these lines.

Other parts of the introduction should also be written more briefly to avoid the reader's boredom. Furthermore, more attention should be paid to the issue of food, packaging, and work surfaces, and the ways of transmission in the food manufacturing systems than to the virus itself and the symptoms of the disease.

Not an attractive title. It's a bit crowded: “2. Persistence and detection of SARS-CoV-2 on food, packaging, and food contact sur- 111 faces, and disinfection methods”.

Line 148, “However, it must be considered that the inoculum used …”; please provide a scale of the numbers of the inoculated rate compared to the natural rate

Line 154, “Van Doremalen et al. (2014) reported that MERS-CoV survived in dromedary camel milk 154 at 4 °C for 72 h…”; if there is any information about the survival rate of SARS-COV-2 in milk, please mention here.

After finishing each part (2.2, 2.3, and 2.4), I would like to read a mini conclusion based on the experience of the authors (according to their knowledge and the result of mentioned literature).

3. Impact of COVID-19 on the food system

This part is well written and tries to cover all related topics and could be used by related researchers.

Part 4. Outbreaks of COVID-19 linked to food production systems

After the literature review, in the last paragraph of this section, please list the limitation and weaknesses of the food system in the face of this pandemic and make suggestions to increase the tolerance and resilience of the system in the face of the next pandemics.

Part 5 and 6:

Please use also the following similar articles to improve the discussion and conclusion part:

1.      Dudek, M.; Spiewak, R. ´ Effects of the COVID-19 Pandemic on Sustainable Food Systems: Lessons Learned for Public Policies? The Case of Poland. Agriculture 2022, 12, 61. https://doi.org/10.3390/ agriculture12010061

2.      Bisoffi S, Ahrné L, Aschemann-Witzel J, Báldi A, Cuhls K, DeClerck F, Duncan J, Hansen HO, Hudson RL, Kohl J, Ruiz B, Siebielec G, Treyer S and Brunori G (2021) COVID-19 and Sustainable Food Systems: What Should We Learn Before the Next Emergency. Front. Sustain. Food Syst. 5:650987. doi: 10.3389/fsufs.2021.650987 (important article for conclusion and solution for future pandemic)

3.      Marta G. Rivera-Ferre, Feliu López-i-Gelats, Federica Ravera, Elisa Oteros-Rozas, Marina di Masso, Rosa Binimelis, Hamid El Bilali, The two-way relationship between food systems and the COVID19 pandemic: causes and consequences, Agricultural Systems, 2021, 103134, ISSN 0308-521X, https://doi.org/10.1016/j.agsy.2021.103134.

For improving the quality of the work, please give some suggestions related to these topics in the last section of the review:

·         Surveillance, quick response teams, and case investigation

·         The positive and negative impacts on global food security

·         The impact of this pandemic on food production, agricultural and fishery/aquaculture supply chains, and markets, in a separate part.

·         What could be the main role in the management of these conditions by organizations such as FAO?

Furthermore, before the conclusion, please list the possible limitations of this work.

Best

Round 2

Reviewer 2 Report

My comments have been addressed.